# Exploring the Potential of siRNA Delivery in Acute Myeloid Leukemia for Therapeutic Silencing

**DOI:** 10.3390/nano13243167

**Published:** 2023-12-18

**Authors:** Anyeld M. Ubeda Gutierrez, K. C. Remant Bahadur, Joseph Brandwein, Hasan Uludağ

**Affiliations:** 1Department of Biomedical Engineering, Faculty of Medicine & Dentistry, University of Alberta, Edmonton, AB T6G 2R3, Canada; 2Department of Chemical & Materials Engineering, Faculty of Engineering, University of Alberta, Edmonton, AB T6G 2R3, Canada; 3Department of Medicine, Faculty of Medicine & Dentistry, University of Alberta, Edmonton, AB T6G 2R3, Canada; 4Faculty of Pharmacy and Pharmaceutical Sciences, University of Alberta, Edmonton, AB T6G 2R3, Canada

**Keywords:** leukemia, siRNA, lipopolymers, nanoparticles, precision medicine

## Abstract

We investigated the feasibility of using siRNA therapy for acute myeloid leukemia (AML) by developing macromolecular carriers that facilitated intracellular delivery of siRNA. The carriers were derived from low-molecular-weight (<2 kDa) polyethyleneimine (PEI) and modified with a range of aliphatic lipids. We identified linoleic acid and lauric acid-modified PEI as optimal carriers for siRNA delivery to AML cell lines KG1 and KG1a, as well as AML patient-derived mononuclear cells. As they have been proven to be potent targets in the treatment of AML, we examined the silencing of *BCL2L12* and *survivin* and showed how it leads to the decrease in proliferation of KG1 and stem-cell-like KG1a cells. By optimizing the transfection schedule, we were able to enhance the effect of the siRNAs on proliferation over a period of 10 days. We additionally showed that with proper modifications of PEI, other genes, including *MAP2K3*, *CDC20*, and *SOD-1*, could be targeted to decrease the proliferation of AML cells. Our studies demonstrated the versatility of siRNA delivery with modified PEI to elicit an effect in leukemic cells that are difficult to transfect, offering an alternative to conventional drugs for more precise and targeted treatment options.

## 1. Introduction

Acute myeloid leukemia (AML) is the most prevalent myeloid disorder in adults [1], with a mortality rate exceeding 90% for patients over the age of 65 [2]. AML originates in hematopoietic stem and progenitor cells, and it is characterized by an increase in immature myeloblasts or “blasts” in circulation, overcrowding the healthy cells. The abnormal growth and differentiation of the myeloblast cell population are driven by mutations and chromosomal alterations that lead to differentiation blocks, which arrest the cells in immature stages of development [3]. The molecular features of AML have been extensively characterized, and now AML is classified as a highly heterogeneous disease from a molecular and pathological point of view [4]. Currently, AML includes around 11 categories of patients depending on sequence analysis and cytogenetics and more than 20 subsets when considering cellular differentiation states [5,6]. A plethora of mutations may lead to the development of the disease, and as a result, multiple molecular targets have been exploited for therapy [7]. While hematopoietic stem cell transplantation (HSCT) allows for the best prevention of AML recurrence and increased disease-free survival, it has the highest treatment-related morbidity and mortality, especially in older patients [8]. The standard chemotherapy treatments had not significantly changed for almost 40 years [9], until the recent introduction of several new therapies: Midostaurin for treating fms-like tyrosine kinase 3 (*FLT3*)-mutated AML, enasidenib for relapsed or refractory AML with an isocitrate dehydrogenase-2 (IDH2) mutation, CPX-351 for newly diagnosed therapy-related patients or with myelodysplasia-related changes, gemtuzumab ozogamicin (GO) for treatment of adults with newly diagnosed CD33+ AML, and ivosidenib for relapsed or refractory AML with an IDH1 mutation [10,11,12,13]. However, most AML patients also become chemo-resistant [14], so new treatment options are likely to be needed, especially for patients who endure high risks with chemotherapy and who are not eligible for HSCT.

Gene silencing by RNA interference (RNAi) aimed at down-regulating or repressing genes that are over-expressed or mutated in AML cells could provide a superior alternative to chemotherapy [15]. While small molecule drugs cannot access every disease-causing protein or aberrant gene, RNAi can be easily implemented to target any disease-causing genetic sequence [16]. RNAi by double-stranded RNA was first discovered in *Caenorhabditis elegans* as a method to control gene expression [17] and was subsequently shown to be active in mammalian cells [18]. The process of RNAi starts with the incorporation of macromolecular short interfering RNA (siRNA) into the effector nuclease, the RNA-induced silencing complex (RISC), where the sense strand is cleaved by Argonaute proteins, and the antisense strand guides it to cleave complementary mRNA sequences [19]. Using siRNA may provide advantages as compared to small molecule inhibitors, which have been shown to be toxic and associated with resistance emergence [20]. The Kiyosawa and Druker groups were amongst the first to test RNAi in AML models. Kiyosawa et al. successfully targeted the Raf-1 kinase and Bcl-2 protein as their over-expression had been involved with chemo-resistance [21]. They showed that siRNA was more effective at achieving gene silencing than antisense oligodeoxynucleotides. Using RNAi to target *FLT3* in AML, an important mediator of survival, proliferation, and differentiation of blasts, siRNA treatment in combination with a small molecule inhibitor was more effective for treatment than either method alone [22]. These studies also revealed the main challenge of delivering siRNAs in a clinical setting. In the case of Kiyosawa, 400 nM of siRNA had to be delivered with nanoparticles composed of a polyamine (Oligofectamine) to achieve the desired silencing. Typically, concentrations in the order of 50–60 nM in culture are considered viable to be translated to a clinical setting, and concentrations of 0.3–0.01 mg/kg have been reported effective in mice and non-human primates with synthetic lipidic carriers [23,24]. In the case of Druker et al., the transfection was performed via electroporation, which is not suitable for in vivo delivery. Merkerova et al. compared some of the chemical transfection methods in chronic myeloid leukemia (CML) models (specifically Oligofectamine, Metafectene, and siPORT Amine) to electroporation and revealed that synthetic carriers usually showed very low delivery for patient cells, and electroporation usually led to high toxicity even though it had a higher capacity for transfection [25]. The transfection of hematopoietic cells especially poses a steep delivery challenge as these cells grow in suspension without attachment. The suspension-growing cells have reduced surface area available to encounter and uptake transfection reagents, and they possess morphological differences that affect surface recognition and penetration of particles [26,27].

Given their ease of synthesis and chemical flexibility with the incorporation of functional groups, we used polymeric macromolecules to implement siRNA delivery [28]. We used low molecular weight polyethyleneimine (PEI) modified with aliphatic lipid groups, which are shown to turn the PEI into an effective delivery system for AML cells [29]. PEI is a cationic polymer that complexes and encapsulates the siRNA into a polyplex nanoparticle via electrostatic interactions [30]. PEI is also known to have a high proton buffering capacity in internalized endosomes, allowing it to bind H^+^ and increase the endosomal osmotic pressure that ultimately ruptures the endosomal membrane [31,32]. Combining polymeric carriers with lipids has been shown to increase the hydrophobicity of siRNA complexes and ease their interactions with hydrophobic cellular membranes, allowing for a more effective cargo delivery to the cytoplasm [26]. Here, we compared different aliphatic groups that are able to improve the delivery features of PEI with a focus on leukemic stem cell models and primary cells. We focused on the genes BCL2 Like 12 (*BCL2L12*) and *survivin* (also known as Baculoviral IAP Repeat Containing 5, *BIRC5*) genes. BCL2L12 is an anti-apoptotic protein that has previously been shown to promote the proliferation of leukemic cells. Upon down-regulating *BCL2L12*, the engraftment of the cells in mice models is impaired [33]. Thomadaki et al. also evaluated *BCL2L12* expression levels in AML patients and compared them to healthy donors; AML patients had increased *BCL2L12* expression that gave them a predisposition for relapse [34]. *BCL2L12* overexpression was also predictive of a shorter overall survival in CML patients [35]. *Survivin* is also part of an inhibitor of the apoptosis family and has been shown to be overexpressed in the majority of cancers [36]. In AML patients, *survivin* is an indicator of poor prognosis, associated with drug resistance, and it is overexpressed in the leukemic stem cell (LSC) population compared to the leukemic population, making it a promising target for down-regulation [37,38]. Its overexpression has been shown to have a role in AML initiation, making it a suitable target in the early stages of treatment [39]. Therefore, this study explored the feasibility of delivering specific siRNAs against these targets using macromolecular carriers to better assess the potential of siRNA in the management of leukemia.

## 2. Materials and Methods

### 2.1. Materials

Low-molecular-weight (0.6, 1.2, and 2.0 kDa) branched PEI, and 25 kDa branched PEI, linoleoyl chloride (LA), stearoyl chloride (StA), lauroyl chloride (Lau), caproyl chloride (CA), alpha-linoleoyl chloride (αLA), and Hanks’ Balanced Salt Solution (HBSS without phenol red) were obtained from Sigma-Aldrich (St. Louis, MO, USA). Lipofectamine^TM^ RNAiMAX was purchased from Invitrogen (Grand Island, NY, USA). Penicillin (1000 U/mL) and streptomycin (10 mg/mL) were purchased from Invitrogen. Fetal bovine serum (FBS) was purchased from VWR (PAA, Ottawa, ON, Canada). RPMI Medium 1640 with L-glutamine was purchased from Qiagen (Huntsville, AL, USA) and IMDM medium with GlutaMax from ThermoFisher (Waltham, MA, USA). Unlabeled, negative control siRNA as well as 6-carboxyfluorescein (FAM)-labeled siRNA were from IDT (Coralville, IA, USA). The following siRNAs were also from IDT: survivin (Cat. No. HSC.RNAI. N001012271.12.1; Sense: rArGrArCrArGrArArUrArGrArGrUrGrArUrArGrGrArArGCG, Antisense: rCrGrCrUrUrCrCrUrArUrCrArCrUrCrUrArUrUrCrUrGrUrCrUrCrC), BCL2L12 (Cat. No. HSS.RNAI. N001040668.12; sequence with manufacturer), cell division cycle 20 homolog (CDC20; Cat. No. HSC.RNAi. N001255.12.1; sequence with manufacturer), and superoxide dismutase 1 (SOD-1; Cat. No. HSC.RNAI.N000454.12; sequence with manufacturer). Mitogen-activated protein kinase kinase 3 (MAP2K3; Cat. No. AM16708; sequence with manufacturer) and ribosomal protein S6 kinase A5 siRNAs (RPS6KA5; Cat. No. AM51334; sequence with manufacturer) were obtained from Ambion (now part of ThermoFisher).

### 2.2. Cell Culture

Suspension cells were cultured with the RPMI medium containing 10% FBS (inactivated at 56 °C for 30 min) and 1% Pen/Strept under normal conditions (37 °C, 5% CO_2_ under humidified atmosphere). KG1A and KG1 cells were obtained from the ATCC (Rockville, MD, USA) and cultured under the same temperature and CO_2_ conditions with Iscove’s Modified Dulbecco’s Medium (IMDM) and 20% FBS. Cell lines were subcultured every 3–4 days and maintained at a density of 1 × 10^5^–1 × 10^6^ cells/mL. Primary cells were obtained frozen from the Canadian Biosample Repository (U. of Alberta) and belonged to AML patients with active disease at diagnosis from the University of Alberta Hospital. Human ethics approval was obtained for the described experiments from the University of Alberta Health Research Ethics Board. Experiments involving primary samples were performed within 24 h of thawing after determining their viability with the Trypan blue staining method. For thawing, cells were transferred dropwise to a tube containing DNase I (100 μg/mL) and incubated for 2–4 min, 5 mL of FBS were then added to the DNase/cell mixture dropwise, and the suspension was distributed into different 1.5 mL tubes and spun down at 200 g for 10 min. The supernatant was removed, and cell pellets were resuspended in IMDM supplemented with GlutaMAX (1×). The medium was also supplemented with 100 ng/mL Stem Cell Factor (SCF), 50 ng/mL FMS-like tyrosine kinase 3 ligand (FLT3L), 20 ng/mL Interleukin-3 (IL-3), 20 ng/mL Granulocyte-Colony Stimulating Factor (G-CSF; Shenandoah Biotechnology; Warwick, PA, USA), and 10^−4^ M of β-mercaptoethanol.

### 2.3. Synthesis of Lipid-Modified PEIs

The PEI-modified polymers were synthesized in-house using methods published before [29,40,41] by grafting aliphatic lipids via N-acylation at different feed ratios. The modifications tested include linoleic acid (LA), alpha-linolenic acid, stearic acid (StA), lauric acid (Lau), and caprylic acid (CA), whose structures are shown in Figure 1. The nomenclature of the polymers is exemplified as follows: PEI0.6-CA4 refers to a 0.6 kDa PEI modified with caprylic acid at a feed ratio of 4 CA per PEI. The composition of the polymers is shown in Figure 1.
nanomaterials-13-03167-sch001_Scheme 1Scheme 1Chemical scheme to prepare the PEI polymers used in this study (top). The list of lipid-substituted polymers, the specific lipid used, and the substitution level (no. of lipids per PEI) are summarized in the table below. PEI0.6, PEI1.2, and PEI2.0 refer to the specific molecular weight of the polymer backbone (0.6, 1.2, and 2.0 kDa).
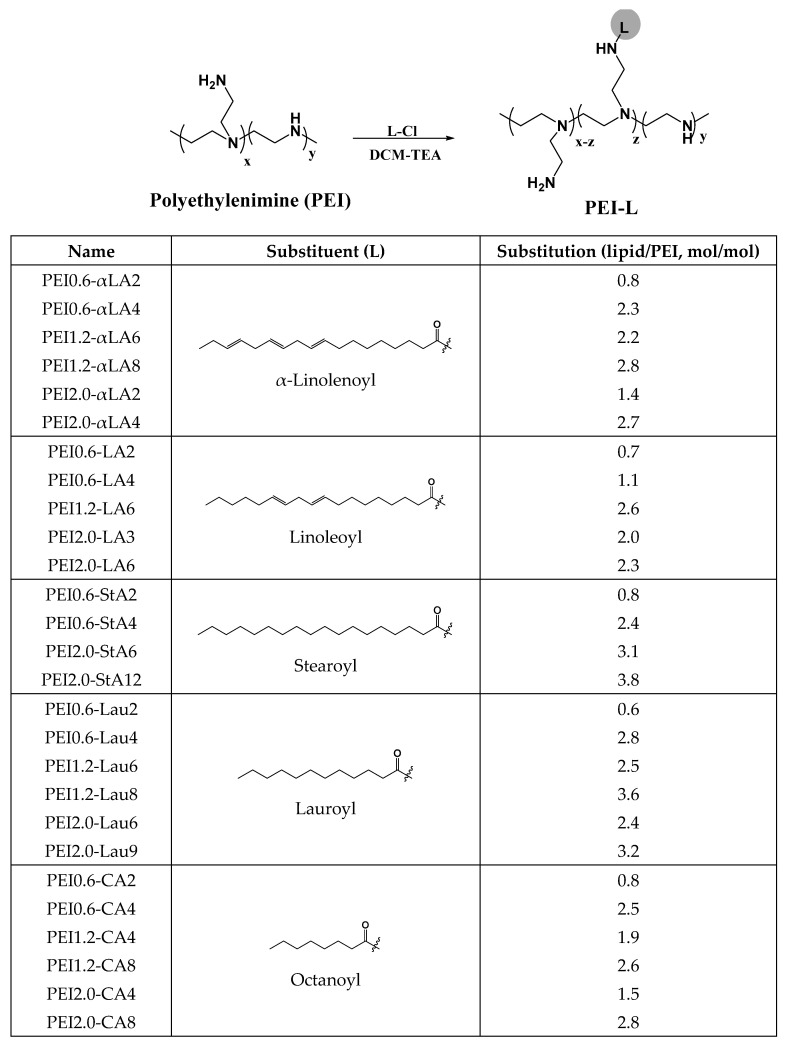


### 2.4. siRNA Delivery

To assess the uptake of the complexes, FAM-labeled siRNA was formulated with a variety of modified carriers. Complexes were prepared by combining siRNA and polymers at different siRNA/polymer weight ratios (1:6, 1:8, and 1:10) and incubating them for 30 min at room temperature in serum-free medium (RPMI or IMDM only) before introducing them to the cells. With RNAiMAX, a weight ratio of 1:2 was used, as increasing this ratio has been shown to be toxic in our previous experience. For delivery experiments, we used an siRNA concentration of 30 nM and approximately 45,000 cells in 48-well plates with a total volume of 400 μL/well after transfection with 100 μL of complexes/well. Cells were analyzed after 24 h of transfections using flow cytometry. On the day of the analysis, the cells were prepared by collecting them in 1.5 mL Eppendorf tubes, centrifuging them at 300× *g* for 5 min, washing them twice with HBSS, and fixing them to a final concentration of 1.2% formaldehyde. All transfections were performed in triplicate. To set the baseline for uptake and fluorescence, a no-treatment control was used, and a gate was set at approximately 1% uptake. For the uptake experiment in Figure 1, we utilized a cell line thought to be THP-1; however, after authentication, they matched the genetic profile of the Raji Burkitt’s Lymphoma cell line. These cells were only used to generate Figure 1 as a model for suspension cells.

### 2.5. Analysis of Cellular Proliferation

The CyQUANT cell proliferation kit was obtained from ThermoFisher, and the manufacturer protocol was followed to assess DNA content. Briefly, cells were transfected on “day 0” and analyzed after 3 days, 7 days, or 10 days after treatment (Figure 2). Cells were also transfected on day 7 for most studies, with the exception of Figure 2. On the day of the analysis, cells were collected in 1.5 mL Eppendorf tubes, centrifuged at 300× *g* for 5 min, rinsed twice with HBSS, and the pellets were lysed using 250 μL of lysis buffer prepared in-house (0.5 mol/L 2-amino-2-methylpropan-1-ol and 0.1% Triton-X; pH 10.5). After 1 h of incubation in the lysis buffer, we prepared a DNA calibration curve including concentrations of 2.0, 1.0, 0.5, 0.25, 0.125, and 0.0 μg/mL of DNA by preparing serial dilutions of the DNA standards provided by the kit in the lysis buffer and adding 1× dye to the samples in a black 96-microwell plate. The fluorescent intensity values were obtained using the Fluoroskan Ascent plate reader (ThermoFisher) with λ_ex_ = 485 and λ_em_ = 530 nm and graphed against the DNA concentrations. The samples were prepared in a similar way by adding 50 uL of the lysed cells and 50 uL of 1× dye to the wells. The DNA concentration from the samples was then obtained using the fluorescent intensity and comparing it to the calibration curve from the standards. The results were plotted as a percentage of DNA by setting the no-treatment control as the reference: [DNA from sample]/[DNA from no-treatment control] × 100. 

### 2.6. Apoptosis Analysis

To analyze the effect of siRNA delivery with the selected polymers, the cells were analyzed for apoptosis using the FITC-Annexin V and Propidium Iodide apoptosis assay kit from BD Biosciences (Cat. No. 556547) following the manufacturer’s instructions. Briefly, the suspension cells were collected in 1.5 mL Eppendorf tubes, centrifuged, and washed with the apoptosis binding buffer (1×) provided. The cells were then incubated with 2.5 μL of FITC-Annexin V and 2.5 μL of Propidium Iodide in the dark for 15 min at room temperature. Then, cells were analyzed with a flow cytometer within 30 min.

### 2.7. mRNA Down-Regulation

To determine the silencing effect of the transfections against a specific gene, RT-qPCR was used to assess the level of intracellular mRNA. First, total RNA from the treated cells was extracted using the TRIZOL method according to the instructions from the manufacturer (Invitrogen, Waltham, MA, USA), and extracts were quantified with a NanoVue spectrophotometer (GE Healthcare, Chicago, IL, USA). For primary cells, cDNA was synthesized using the SuperScript™ VILO™ cDNA Synthesis Kit from ThermoFisher. Otherwise, we used the Invitrogen cDNA kit, including Master Mix 1 (Oligo dT 0.5 μg/uL, random hexamers and dNTPS 10 mM) and Master Mix 2 (5× Synthesis Buffer, DTT 0.1M, RNAout 1.8 U/μL and M-MLV RT enzyme). After adding Master Mix 1 to the samples (2000 ng of RNA), they were heated to 65 °C for 5 min. After adding Master Mix 2, the samples were heated at 25 °C for 10 min, 37 °C for 50 min, and 70 °C for 15 min, and then stored at 4 °C. For the quantitative PCR (see Table 1 for probed used), human beta-actin was used as the endogenous control (forward: 5′-GCGAGAAGATGACCCAGAT-3′ and reverse: 5′-CCAGTGGTACGGCCAGA-3′). An amount of 5 μL of master mix containing 2× SYBR Green (FroggaBio Cat. No. BIO-92005, otherwise obtained from the Molecular Biology Facility MBSU at the U. of Alberta) and 1.0 μL of each forward and reverse primer (10 uM) per sample were combined and added to 3 μL of cDNA (7.5 ng/μL). The samples were analyzed using a StepOne Real-Time PCR System (Applied Biosystems; Foster City, CA, USA) based on the manufacturer’s recommendations (initial denaturation for 10 min at 95 °C, followed by 40 cycles of denaturation at 95 °C for 15 s and hybridization and elongation at 60 °C for 1 min); the results were then processed using the 2^−ΔΔCT^ method and presented as relative quantities normalized to the beta-actin housekeeping gene.

### 2.8. Statistical Analysis

All results were plotted as means of 3 replicate samples with standard deviations. Statistical significance was determined using the unpaired Student’s *t*-test or ANOVA with Tukey HSD where an asterisk (*) represents (*p* ≤ 0.05) and a plus sign (+) represents (*p* ≤ 0.1). The analysis was performed by comparing the control siRNA (csiRNA) sample with the siRNA treatment samples.

## 3. Results

### 3.1. Selection of Carriers for siRNA Delivery in Cell Lines

A library of PEIs modified with saturated and unsaturated lipids was analyzed for FAM-siRNA delivery. We determined the percentage of cells displaying uptake and mean fluorescence/cell in Raji Burkitt cells (Figure 1; polymer/siRNA *w*/*w* ratio of 10:1), KG1 (Figure 2A; polymer/siRNA *w*/*w* ratios of 10:1, 8:1, and 6:1), and KG1a cells (Figure 2B; polymer/siRNA *w*/*w* ratios of 10:1, 8:1, and 6:1). The RNAiMAX and PEI25 were used as commercial reagents for comparison with in-house prepared PEIs. Based on the overall results, PEI1.2-Lau8 was consistently the most effective carrier with FAM-siRNA delivery up to ~95% of cells and with the highest fluorescence/cell in the cell lines. The PEI1.2-LA6 was also relatively effective in our screens, in line with previous studies that showed its effectiveness for delivery of siRNA to AML cells [42,43] (in addition to breast cancer MDA-MB-231 cells [44]). Other polymers effective in KG1 and KG1a cells and selected for further experiments were PEI0.6-Lau4, PEI1.2-St4, and PEI2-LA9. The LA substitution was the only unsaturated lipid modification successful in comparison to other substituents. We had previously hypothesized that increased unsaturation gave the carrier more fluidity, which allowed for better interactions with the cell membrane [29]. In general, we observed that the delivery efficiency was increased with increased substitution levels (Figure 1C,D). Our studies showed that PEI25 was not effective at all in Raji Burkitt cells as well as the starting PEIs (0.6 to 2.0 kDa) used in our synthesis, but RNAiMAX was quite effective in siRNA delivery, giving as good delivery as the most effective in-house synthesized polymers.

### 3.2. Growth Inhibition by siRNA Delivery in Cell Lines

We first examined siRNA-mediated growth inhibition in Raji Burkitt cells. One siRNA treatment was undertaken on day 0, and cellular proliferation was assayed on days 7, 10, and 15 using the commercial RNAiMAX and PEI2.0-LA6 as delivery agents. A range of relevant siRNAs were used that were previously employed in our lab to prevent the proliferation of malignant cells; no obvious growth inhibition was seen at the end of the experiment for both delivery systems. We repeated this study by using two siRNA treatments (on days 3 and 7) and analyzed the cell proliferation over a period of 10 days. Growth inhibition was not observed on day 3 with most siRNAs, except with BCL2L12 siRNA delivered with PEI1.2-LA6 (~14% inhibition, Figure 3). After the second siRNA treatment on day 7, up to 80% of growth inhibition was seen by targeting RPS6K5A with PEI1.2-LA6. By delivering two doses of siRNA on days 0 and 7, most siRNAs were able to reduce proliferation by at least 40% with PEI1.2-LA6, with the exceptions of CD29 and PIK3CB. The PEI1.2-LA6 carrier was more effective at inhibiting growth on day 10 than the RNAiMAX under these conditions.

Using a selection of groups of specific siRNAs (BCL2L12, CDC20, MAP2K3, RPS6KA5, SOD-1, and survivin) from the first screen, KG1a cells were then treated with siRNA complexes to decrease cell proliferation (Figure 4). Six carriers with the highest siRNA delivery from Figure 1 and Figure 2 were employed to identify the optimal carrier, including RNAiMAX, PEI1.2-LA6, PEI2-LA9, PEI1.-St4, PEI1.2-Lau4, and PEI1.2-Lau8. The PEI2-LA9, for reasons not clear at this stage, gave increased proliferation with CDC20, SOD-1, and survivin siRNAs, and thus was not used for further studies (Figure 4). In terms of growth inhibition, PEI1.2-St4 did not show any significant effect, but PEI1.2-Lau8 was the most successful carrier among modified PEIs at significantly inhibiting cell growth with four out of six specific siRNAs on day 10. Most inhibition was seen after targeting MAP2K3 with five out of six carriers on day 10, SOD-1 with four out of six carriers, and BCL2L12 with three out of six carriers (Figure 4). In KG1 cells, siRNA delivery was undertaken by using RNAiMAX, PEI1.2-LA6, and 1.2PEI-Lau8. For these cells, control siRNA impaired cellular proliferation on day 3 (indicating non-specific toxic effects), and increased cell proliferation was observed with different targets for the carriers instead of the desired growth inhibition. Growth inhibition was observed for delivery of survivin siRNA with PEI1.2-LA6 on day 10 and MAP2K3 siRNA with PEI1.2-Lau8 on day 7 (Figure 5).

As we observed some toxicity with control siRNA complexes in both KG1a and KG1 cells at times, lower amounts of modified PEIs were used in complexes (polymer/siRNA ratio of 6:1) to assess if decreased polymer amount would eliminate the toxic effects. Only siRNAs targeting BCL2L12 and survivin were employed for this purpose (Figure 6). In both KG1 and KG1a cells, the growth inhibition observed with PEI1.2-LA6 and control siRNA at a ratio of 10:1 was mitigated by lowering the ratio to 6:1. Specific combination of polymeric carriers and siRNAs led to growth inhibition on day 7 and day 10 in KG1 cells, in particular PEI-1.2-LA6 and PEI1.2-Lau8 and BCL2L12 siRNA combinations. In KG1a cells, BCL2L12 siRNA decreased the growth on day 10 with PEI1.2-Lau8 and PEI0.6-Lau4 by ~30%. Survivin siRNA decreased the growth on day 10 with PEI1.2-LA6 but only by ~10%. In contrast, survivin siRNA was more effective in reducing cell growth on day 3 with the three carriers, giving ~20% growth inhibition on average (Figure 6), but this did not reach significance (*p* < 0.1).

### 3.3. mRNA Silencing and Apoptosis in KG1a and KG1 Cells

We used qPCR to assess the effect of siRNA delivery at the mRNA level. We analyzed the targets BCL2L12, survivin, and MAP2K3 in KG1a cells at the PEI/siRNA ratio of 6:1 (Figure 7A). The BCL2L12 was significantly silenced with all carriers, MAP2K3 was silenced with PEI1.2-LA6, and the survivin with PEI1.2-LA6 and PEI1.2-Lau8. In contrast, MAP2K3 was up-regulated with PEI0.6-Lau4. In KG1 cells (Figure 7B), PEI1.2-LA6 was the most effective at silencing survivin. The PEI1.2-Lau8 also achieved some silencing with BCL2C12 and surviving siRNAs, but PEI0.6-Lau4 did not have any effect on mRNA levels (Figure 7B), even though it was one of the more effective siRNA delivery agents for KG1 cells (see Figure 2).

We then examined the levels of apoptosis and cell death in KG1a cells after BCL2L12, MAP2K3, and survivin siRNA delivery using PEI1.2-LA6, PEI0.6-Lau4, and PEI1.2-Lau8 (a ratio of 6:1). Annexin V and PI staining were used on day 3 to determine the percentage of early apoptotic (annexin V+/PI−) or late apoptotic/dead cells (annexin V+/PI+). Typical flow cytometry histograms are shown in Appendix A. Delivery of control siRNA did not lead to an observable change in early and late apoptotic levels (Figure 8). The siRNA delivery with PEI1.2-LA6 increased early apoptosis with all three targets, while PEI0.6-Lau4 increased early apoptosis when targeting survivin alone. The late apoptosis and cellular death were promoted by targeting MAP2K3 with PEI1.2-Lau8 (Figure 8).

### 3.4. siRNA Delivery and Silencing in AML Patient Cells

We again conducted a screen of in-house prepared PEIs for siRNA delivery to patient cells since we did not want to assume that the polymers identified from cell line studies were optimal for patient cells as well. Among the commercial carriers, RNAiMAX was relatively more effective than PEI25 in siRNA delivery (Figure 9). However, RNAiMAX gave a relatively lower percentage of siRNA delivery (2% to 33%), while the PEI1.2-Lau8 gave a higher percentage of siRNA-positive cells (3% to 87%), while the mean uptake was equivalent for both carriers in the select patient. Patient-to-patient variability in siRNA delivery was most evident for best-performing reagents; while patient 4 gave little uptake with both reagents, other patients displayed variable siRNA uptake for both reagents and AML patient cells (Figure 9). From the patient studies, PEI0.6-St4 was additionally selected as it consistently delivered FAM-siRNA at higher percentages in 3/5 patients (Figure 9).

Four patient samples were analyzed for silencing using optimal carriers from the uptake study in Figure 9. The RQ of the targeted mRNAs was first presented as a ratio of specific/control siRNA treatment, using the targets BCL2L12, CDC20, survivin, and RPS6K5A (Figure 10A). Due to limited sample volumes, not all samples were available for various polymer/siRNA treatments (see Figure 10 for the number of replicates, n). The siRNA delivery with PEI1.2-LA6 was able to approximately halve the mRNA transcript level of *BCL2L12* in one patient tested. The survivin was down-regulated consistently with the PEI1.2-Lau8 (n = 4, *p* < 0.05 vs. CDC20 but not RPS6K5A; Figure 10A). The results were also analyzed based on the ratio of control siRNA to no treatment to assess whether control siRNA affected the expression levels of the selected targets (Figure 10B); BCL2L12 and survivin mRNA levels were not affected by this analysis, but PEI1.2-LA6 gave a reduced expression for CDC20 and RPS6K56A, indicating some non-specific effects with this polymer and these specific targets.

## 4. Discussion

The main challenge with the current AML therapies is the lack of complete remission in 10–40% of patients. More precisely, their blast count does not go below 5% after 1 or 2 cycles of induction therapy and is therefore categorized as resistant or primary refractory [45,46]. Most of these patients are then advised to reduce the disease burden before considering an HSCT [47]. Over the years, LSCs have been identified as the main culprit for the progression of leukemia as well as for patient relapse. Their self-renewal capacity, differentiation potential, and effector functions allow LSCs to maintain and regulate the disease, leading to resistance to chemotherapy and other targeted therapies [48]. As a result, focusing on LSCs is one of the most promising approaches to tackle chemo-resistance and decrease disease morbidity. For this reason, we have focused these studies on the identification of siRNA delivery systems to KG1 and KG1a stem cell models for AML that can also be translated to primary patient samples. KG1a have been found to keep their self-renewal potential and be inherently resistant to chemotherapy and drug treatments, including daunorubicin- and mitoxantrone-induced apoptosis [49,50], TNFα [51], and natural killer cell killing [52].

This study focused on the screening of PEI-modified polymers to curb the growth of leukemic cells. We previously characterized the size of some of the complexes used in this study, and the hydrodynamic size has been reported to be between 100 and 200 nm after complexation [29,41] so these characterization studies were not repeated here. We demonstrated that PEI1.2-Lau8 was the most effective carrier for all cell lines as well as for primary samples (Figure 1, Figure 2 and Figure 3). This is promising as this chemically modified PEI achieved delivery in up to 80% of KG1a cells, as well as delivery of up to 80% in one of the patient samples with elevated MFI (Figure 2 and Figure 3). Lauric acid is a medium-length saturated fatty acid containing 12 carbons, which is naturally sourced and metabolized [53]. In addition, this fatty acid has been found to have anti-proliferative and pro-apoptotic effects in some cancer cells [54]. The use of lipids for drug delivery, specifically in nano-carrier formulations, is beneficial as they increase membrane interactions and permeability, promoting high levels of cellular uptake, with liposome delivery systems being the first to be translated into clinical applications [55]. In our case, we used lipids to increase the affinity between PEI/siRNA particles and the plasma membrane [56]. However, when using lipid formulations, we must also consider that lipids are highly involved in cell signaling pathways, gene expression regulation, apoptosis, metabolism, and inflammation, amongst other essential cellular response effects [57,58]. Therefore, it is expected to observe some non-specific effects when studying lipid-based carriers as their interactions with the cellular membrane might trigger signaling cascades that might be independent of the target of interest. Such effects might be over-expressed in in vitro studies as the effective concentration of each carrier available to interact with each cell would be significantly greater than its bioavailability in vivo. We noticed that our control treatments promoted cellular proliferation at times (Figure 5) or led to an increase in transcript levels of targeted genes (e.g., BCL2L12 and survivin levels with higher ratio siRNA/PEI complexes; Figure 8). These effects might not only be caused by increased signaling from the lipids themselves but also by endocytotic effects on proliferation, which have been observed before via the Wnt/β-catenin pathway [59,60,61]. We noted that the effects were more prominent with the carriers that had higher levels of lipid substitutions, PEI2-LA9 and PEI1.2-Lau8, and were decreased when the PEI/siRNA ratio was lowered to 6:1, which is why we hypothesize that lipids themselves might be triggering cellular signals that are yet to be explored. Another reason for non-specific effects could be from the control siRNA, as these could arise when siRNA interacts with other mRNA or non-specific sequences, yet these can be reduced with ease by designing different siRNA sequences or by chemically modifying the siRNA [62,63,64].

Focusing on specific effects on cell proliferation, we compared growth inhibition over a 2-week period and showed that maximum effects are seen on day 7 since the outcome of siRNA knockdown is known to be transient, especially as unaffected cells proliferate (Figure 4). However, proliferation was more successfully inhibited by reinforcing the initial treatment after 7 days. By delivering siRNA twice, we were able to induce growth inhibition by up to 80% and by at least 40% with all targets used. We applied the same treatment schedule to KG1 and KG1a cells; for KG1A cells, we were able to achieve up to 45% growth inhibition when targeting MAP2K3 and were able to decrease the proliferation of all targets tested with different carriers with the exception of survivin (Figure 5). When testing the same transfection schedule on KG1 cells, we observed that they were less responsive, yet this was expected as they also had less cellular uptake of siRNA than the KG1a cells (Figure 2). Most of the effects were achieved on day 3, and only by targeting survivin with PEI1.2-LA6 were we able to obtain about 40% decreased proliferation by day 10. More effective carriers might be needed for transfecting KG1 cells. In this instance, it is important to note that the investigated carriers still hold high specificity for cell type, and their effectiveness changes even when comparing two similar cells that share the same origin. Since our carriers do not bear any targeting cell-surface ligands for binding, this highlights the importance of the lipid membrane interactions and the specific lipids present in the carriers. To reduce some of the non-specific effects, we lowered the PEI/siRNA ratio and noticed that the KG1a cells were still more responsive to the siRNA than KG1 cells even though the effects were not as pronounced as using the 10:1 ratio. For both cells, PEI1.2-Lau8 was the most effective carrier, and both BCL2L12 and survivin were able to inhibit proliferation with effects lasting up to 10 days in KG1A cells (Figure 7). To examine the outcomes of siRNA targeting at the molecular level, we analyzed the mRNA transcript levels using RT-qPCR. We were able to achieve up to 40% reduction in mRNA content of the KG1a cells with the ratio of 6:1 when comparing the control siRNA to the treatment. This siRNA treatment was also tested in KG1 cells, and the most transcript suppression was achieved by using PEI1.2-LA6 and survivin siRNA, obtaining ~20% down-regulation (Figure 9). In the past, we observed that the PEI/siRNA ratio is critical for the intracellular dissociation of siRNA from the carrier to happen more readily, and this could lead to more effective mRNA down-regulation. Finally, we analyzed siRNA silencing on primary cells by targeting BCL2L12, CDC20, survivin, and RPS6K5A. Based on mRNA changes in specific vs. control siRNA treatments, we observed a significant elevation of BCL2L12 levels in two patients with PEI1.2-Lau8, indicating an active up-regulation of mRNA levels by targeting this anti-apoptotic protein with this delivery system. Compared to CDC20, survivin silencing was also obtained using this polymer, but the large variation in RPS6K5A levels hindered any firm conclusions. Despite significant variations in siRNA delivery among the patient cells, we showed that survivin was significantly down-regulated without any significant changes in its level with a control (scrambled) siRNA treatment. The reasons for patient-to-patient variations in siRNA delivery are not known and will be addressed in future studies.

As we were examining targets that are involved in the regulation of proliferation and apoptosis, Annexin V/PI staining was inspected in KG1a cells, and we observed how PEI1.2-LA6 was the most effective carrier to increase apoptosis with all targets tested. However, survivin with PEI0.6-Lau4 increased early apoptosis the most, and MAP2K3 with PEI1.2-Lau8 was the most effective at promoting cellular death. With this assay, it appeared that each target behaved differently depending on the choice of carrier, again highlighting the importance of carriers to contribute to cellular outcomes. For a more thorough analysis of how silencing influences apoptosis, more time points need to be studied to obtain a better understanding of the peak apoptotic effect. In this way, we can also design a treatment regime that allows for the reinforcement of apoptotic effects along with the optimized anti-proliferative event.

This study highlighted the versatility of the polymeric carriers modified with aliphatic lipid groups. By using different PEIs and lipids, we have access to countless chemical modifications that can be screened for individual patient samples to create personalized delivery systems. Lipid-modified polymers for nucleic acid delivery were described almost 2 decades ago, but most of the early work was concentrated on DNA delivery [65,66], including our work [67]. With the emergence of RNAi technology in the mid-2000s, lipopolymeric materials were adopted for siRNA delivery, and our own work showed the feasibility of this approach in leukemia models [68]. This study explored the effect of different lipid substitutions on the obtained anti-leukemia efficacy in vitro using a range of oncotargets, which were implicated in both leukemia and other cancers and were not explored with the described approach. It is possible to design and optimize the siRNA target, given the molecular information about the patient’s individual transcriptome, so that personalized therapy can be undertaken. For RNAi use in clinics, we propose that it can serve as an alternative for patients who do not respond to induction therapy or who are older and more vulnerable to stand-alone chemotherapy. RNAi could be paired up with the current therapies to allow for a lower dosage of the drugs. In this initial exploratory study, we observed that not all targets gave the same level of mRNA reduction, or the resultant silencing did not always correlate with a reduction in proliferation or increase in apoptosis. A better insight into the effect of mRNA down-regulation could be obtained by expanding the time point analysis of proliferation/apoptosis, direct analysis of protein down-regulation, and induction of compensatory signaling events. Correlating mRNA levels with protein levels has also not always been successful [69], although our experience with breast cancer models has shown a good correlation when siRNA was delivered with modified PEIs [44]. Some genes have also shown less variation in mRNA levels during cellular division, and instead, their protein levels are controlled at the post-translational level [70]. To overcome these limitations, siRNA targeting could be analyzed across a group of genes in the same cellular pathway as each one might have variations in the above processes yet still allow for similar outcomes in lowering cellular proliferation, inducing apoptosis or differentiation. Combinational targeting of siRNAs against multiple targets could also be employed in the future to study synergistic effects of complementary therapy, for example, targeting the Ras/MAPK pathway involved in cellular differentiation, growth, chemotaxis and apoptosis, and the BCL-2 intrinsic apoptotic pathway.

## 5. Conclusions

To explore alternative treatments for the AML disease, we investigated the feasibility of using macromolecular siRNA-based therapy for the disease by developing effective macromolecular (polymeric) carriers that facilitated intracellular delivery of the siRNA. Optimal lipid-substituted polymers were identified that provided effective delivery in AML cell lines KG1 and KG1a, as well as AML patient-derived mononuclear cells. As they have been proven to be potent targets in the treatment of AML, we examined the silencing of both *BCL2L12* and *survivin* and showed how it leads to the decrease in proliferation of the KG1 and KG1a leukemic stem cell models at different time points. By optimizing the transfection schedule, we were able to enhance the effect of the siRNAs on proliferation over a period of 10 days. We additionally showed that, with proper modifications for the low molecular weight PEI, multiple other genes, including *MAP2K3*, *CDC20*, and *SOD-1*, could be targeted to decrease the proliferation of the employed cells. The overall studies in this work demonstrated the versatility of siRNA delivery with modified PEI to elicit an effect in leukemic cells that are difficult to transfect as well as paramount therapeutic targets for AML, offering an alternative to conventional drugs for more precise and targeted treatment options.

## Data Availability

Original data is available from the authors upon request.

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
