# Peer review of "Exploring the Potential of siRNA Delivery in Acute Myeloid Leukemia for Therapeutic Silencing"

_nanomaterials, 2023, doi:10.3390/nano13243167_

Round 1
Reviewer 1 Report
Comments and Suggestions for Authors
The authors tested the feasibility of their carrier for siRNA delivery against AML cells. The in vitro studies were performed and they found the optimal carrier. A major concern was the applicability to the in vivo use.
1. I suppose the carrier may be accumulated in the lung or liver due to the absence of an active mechanism for prolonged systemic circulation.
2. Taking the solution composition into account, the particle size of the lipid-PEI/siRNA complexes may be large. I think such large complexes may not be suitable for in vivo use.
3. The siRNA should be delivered to bone marrow. Large particles may not be delivered to bone marrow.
4. The authors did not pay attention to the reproducibility of the results in other research groups. They should write the method section in more detail.
5. About siRNA: polymer ratios, it was unclear whether the ratio was weight ratio, molar ratio, or NP ratio.
6. The composition of the transfection medium was unclear. Serum-free?
7. The description for the PEI25k preparation condition was missing in the method section.
8. siRNA sequences were unclear.
9. The toxicities of the carriers were unclear.
10. Tick labels for the horizontal axis were missing in Fig. 1A.
11. Rpm for centrifuge should be converted into centrifugal force.
12. Supplementary figures were missing for peer review.
13. Unusual abbreviations (SCF, FLT3L, etc.) should be spelled out.
14. There were statistical errors. The authors should use multiple comparison tests such as Tukey’s test.
Reviewer 2 Report
Comments and Suggestions for Authors
The manuscript by Ubeda Gutierrez et al. describes an interesting piece of work on siRNA carriers and potential use for therapeutic silencing in acute myeloid leukemia. However, I would suggest the following:
Materials. " linoleyl chloride (LA)": The authors probably mean linoleoyl chloride or linoleic acid. Please check. "caprylic chloride": caproyl chloride or caprylic acid? " alpha-linoleyl acid ( LA)": alpha-linoleic acid or alpha-linoleoyl acid?
Scheme 1. The structure of alpha-linolenoyl and linolenoyl is not correct. Double bonds must have the cis configuration, not the trans one. Did the authors check that acylations exclusively took place at primary amines as indicated on the reaction scheme? Otherwise, the scheme should be modified or this point commented.
Figure 1A-D. The number of bars in fig. 1A and 1B is different (29 vs 30) so fig 1A cannot be interpreted. The legend does not properly explicit Fig. 1C and 1D? What are the symbols for? My own experience is that, unexpectedly, part of siRNA modified with a fluorophore may enter cells without the need for a carrier. Did the authors check this point? As a control, cells should be treated with naked siRNA to verify that uptake and MFI are indeed close to zero or not significant. As siRNA do not generally form stable complexes with low molecular weight PEI and can readily dissociate, interpretation of the data could be misleading in the absence of this control. This holds as well for fig. 9.
Figure 2. What are 6/8/10 for? PEI/siRNA ratios? w/w ratios? Molar ratios? Do please indicate in the figure legend.
Figure 3. Please label RPS6K5A instead of RPS and precise what the carrier:siRNA ratio corresponds to (w/w or mol/mol)?
General.
1-Resolution of the figures should be improved.
2-SI section (Fig. S1) could not be found.
3-A number of PEI-derived lipopolymers for nucleic acid delivery have been already described in the literature. The authors might refer to these previous reports (e.g., Bioconjugate Chem 2001, 12:337. Biomacromol 2002, 3:1197. J Gene Med 2011, 13:46. Adv Genet. 2014, 88:263…) and discuss about the originality of their own work.
Concluding comment.
Efficiency of intracellular delivery of nucleic acid depends on multiple factors of which, and far from least, the structure of the transfection reagents, lipoplexes, polyplexes or lipopolyplexes. Among the key parameters are thus the size and ionization state (e.g., zeta potential) of the transfection particles. However, these two parameters have not been examined in this work that definitely suffers from this lack. Interpretation of the collected data would most probably benefit from a precise characterization of the lipopolyplexes. What is their electrostatic charge? Does it allow electrostatic interaction with the negatively charged cell membrane? Do the lipopolymers self assemble through hydrophobic interactions between the aliphatic chains or not? In other words, do they aggregate? Do they sediment well on the cell layer (the larger the better they sediment)? All these questions should be answered before further consideration for publication.
Reviewer 3 Report
Comments and Suggestions for Authors
The authors used siRNAs directed against BCL2L12 and survivin transfected into AML cells and AML patient-derived mononuclear cells with the aid of specific newly developed PEI modified with fatty acids such as lauric and linoleic acids (among others). They observed a high transfection efficiency and inhibition of growth of AML cells, which are difficult to transfect very specially when using the PEI1.2-Lau8.
The choice of the target genes is appropriate since they are genes codeing for antiapoptotic protien, thus their inhibition leads to apoptosis and cell death. The work has also the add value of showing results of experiments performed with patient samples.
Suggestions:
- Is it possible to indicate the sequence/s of siRNAs used against the selected targets (Bcl2L13, survivin and so on), or they cannot be disclosed as they were purchased from IDT ?. If the sequences are known they could be shown in a table.
- Described the characteristic of the cell lines used, KG1 and KG1a.
Minor:
- In the abstract I suggest to use the word "modified" instead of substituted (line number 15, third of the abstract) for clarity.
- Line 140, centrifugue force should be writen 14,000 rpm
- specify in fig 7 and 8, the meaning of the blue color. NT (not transfected ?)
Round 2
Reviewer 1 Report
Comments and Suggestions for Authors
The responses from the authors were not satisfactory. They did not reply to my comments sincerely. The most important issue "reproducibility" was not resolved. For example, I failed to search the siRNAs on the IDT homepage.
Author Response
I will disagree with the reviewer's comments. We did our best to respond to the comments. The one point highlighted is very specific and do indeed show that we responded with an exact answer. We articulate the exact compounds bought from IDT with specific catalogue numbers. It is possible that (i) the reviewer could not find them on the IDT web site due to hardship in navigation, or (ii) the IDT changed their catalogue numbers or discontinued perhaps. We contacted IDT today and asked them about their products. Below is the response:
"Good afternoon, Thank you for your email. While HSC.RNAI. N001012271.12.1 can't be ordered by pulling the design from our website, I've provided the sequences below.
Sense: rArGrArCrArGrArArUrArGrArGrUrGrArUrArGrGrArArGCG
Antisense: rCrGrCrUrUrCrCrUrArUrCrArCrUrCrUrArUrUrCrUrGrUrCrUrCrC
Using these sequences, you'll be able to order HSC.RNAI. N001012271.12.1 as a custom DsiRNA duplex here. To order, copy/paste the sequences above into the sequence entry boxes on the ordering page (1 sequence per sequence entry box)."
So, as you can see, anyone can purchase these siRNAs if they desire.
Thank you.
Hasan
(we also note that we published 2 papers before with this siRNA in the context of breast cancer with the same cat no provided in the papers)
Reviewer 2 Report
Comments and Suggestions for Authors
The answers to my questions are acceptable.
Author Response
Thank you for your time and effort in reviewing our submission.